# Preferences for oral PrEP dosing among adolescent boys and young men in three sub-Saharan African countries

**Ayoub Kakande**[1]*, **Andrew Sentoogo Ssemata**[1,2], **Richard Muhumuza**[1], **Millicent Atujuna**[3], **Andrew Abaasa**[1,4], **Denis Ndekezi**[1], **Gugulethu Tshabalala**[5], **Teacler Nematadzira**[6], **Stefanie Hornschuh**[5], **Mangxilana Nomvuyo**[5], **Nadia Ahmed**[3], **Mamakiri Maluadzi**[5], **Helen Anne Weiss**[4], **Emily Webb**[4], **Lynda Stranix-Chibanda**[6,7], **Janan Janine Dietrich**[5,8], **Janet Seeley**[1,2], **Julie Fox**[9]

1 Medical Research Council/Uganda Virus Research Institute and London School of Hygiene & Tropical Medicine Uganda Research Unit, Entebbe, Uganda, 2 Department of Global Health and Development, London School of Hygiene & Tropical Medicine, London, United Kingdom, 3 Desmond Tutu HIV Foundation, Faculty of Health Sciences, University of Cape Town, Cape Town, South Africa, 4 Department of Infectious Disease Epidemiology, London School of Hygiene & Tropical Medicine, London, United Kingdom, 5 Perinatal HIV Research Unit, School of Clinical Medicine, Faculty of Health Sciences, University of the Witwatersrand, Johannesburg, South Africa, 6 Clinical Trials Research Centre, University of Zimbabwe, Harare, Zimbabwe, 7 Child and Adolescent Health Unit, Faculty of Medicine and Health Sciences, University of Zimbabwe, Harare, Zimbabwe, 8 Health Systems Research Unit, South African Medical Research Council, Bellville, South Africa, 9 Department of Infectious Diseases, King's College London, London, United Kingdom

* Ayoub.Kakande@mrcuganda.org

## Abstract

### Background

HIV remains a leading contributor to the disease burden in sub-Saharan Africa, with adolescents and young people disproportionately affected. Optimising pre-exposure prophylaxis (PrEP) uptake has predominantly focused on women and adult men who have sex with men. We explore adolescent boys and young men's PrEP uptake preferences in South Africa, Uganda, and Zimbabwe.

### Methods

A cross-sectional sequential exploratory mixed-methods study amongst males aged 13–24 years was conducted between April and September 2019 as part of the CHAPS trial. Group discussions (GDs) and In-Depth Interviews (IDIs) focused on motivations and hindrances for HIV testing, PrEP preference, and reasons for the uptake of PrEP. A thematic approach was used to analyse the qualitative data. A quantitative survey following the qualitative work covered questions on demographics, HIV risk and PrEP preferences (on-demand vs. daily). For quantitative analysis, we fitted logistic regression models to determine factors associated with on-demand vs daily PrEP preference.

### Results

Overall, 647 adolescent boys and young men (median age 20, IQR: 17–22) were enrolled. Of these, 422 (65.22%) preferred on-demand PrEP (South Africa 45.45%, Uganda 76.80%,

**Data Availability Statement:** All relevant data are available at: https://doi.org/10.17037/DATA.00003232. Quantitative data has been made

publicly accessible. The qualitative data is available upon request due to potentially identifying information. The senior data Manager who will be responsible for ensuring data access is: Jane Frances Lunkuse Jane.Lunkuse@mrcuganda.org +256705402597 Senior Data Manager MRC/UVRI & LSHTM.

**Funding:** This project is part of the EDCTP2 programme supported by the European Union grant number RIA2016MC-1616 CHAPS). The funders played no role in the study design and will not play any role in the data collection, analysis or interpretation, nor in the writing of any manuscripts which may follow these studies.

**Competing interests:** The authors have declared that no competing interests exist.

Zimbabwe 70.35%; p<0.001). Factors independently associated with on-demand PrEP included country (South Africa, adjusted odds ratio (aOR) = 0.19 [95%CI:0.1–0.3] compared to Uganda) and advanced planning of sex [>24 hours in advance aOR = 1.4 (0.9–2.3) compared to <2 hours]. Qualitatively, participants commonly believed they were not at risk of HIV acquisition most of the time and thought that on-demand PrEP would be suitable as they tend to plan sexual activity in advance.

## Conclusion

Preference for on-demand PrEP is high in young males. The qualitative data support a preference for on-demand PrEP in those who plan sex in advance. HIV intervention programs should offer both on-demand and daily PrEP to engage more adolescent boys and young men in HIV prevention practices.

## Introduction

HIV is a leading contributor to the disease burden in sub-Saharan Africa (SSA) with adolescents and young people still disproportionately affected [1]. As of 2020, 4000 new HIV infections (adults and children) were reported daily, of which 60% were from SSA alone, and 31% were among young people (15–24) [2]. The majority of challenges associated with young male HIV acquisition are attributable to the lack of access to HIV prevention, and poor adherence and retention in those who do access HIV prevention [1, 3–5].

In sub-Saharan Africa, the overall prevalence of 'any substance use' is estimated at 41.6% with caffeine containing products at 41.2% and alcohol at 32.8% calling for interventions that are comprehensive and targeted at adolescents these settings [6]. Many adolescents are at risk of HIV because of multiple partnerships and insufficient condom [7].

Oral Pre-exposure prophylaxis (PrEP) is a highly effective form of HIV prevention [8], taken daily or around the time of sexual activity via an on-demand regimen [9]. Oral PrEP has been recommended for people at substantial risk of HIV by the World Health Organisation (WHO) since 2015 [10, 11]. Similarly, on-demand oral PrEP which involves taking oral PrEP around the time of sexual activity (two doses taken 24 hours before sex followed by single doses for the two days after sex) and has been shown to be highly effective in men who have sex with men (MSM) [9]. However, the widespread use of on-demand oral PrEP is hindered by a lack of data in other populations and concerns regarding regimen complexity [11]. Even though the WHO recommends oral PrEP in populations with substantial risk such as MSM [12], other populations such as adolescents and young people are not catered for, suggesting a lack of policies, availability, uptake, or adherence to PrEP [1, 9]. This could be attributed to the fact that this vulnerable group continue to be neglected in low-resource settings where there is lack of specific intervention components and outcome measurements to adequately map these interventions [13].

The sustained high incidence and prevalence of HIV among adolescent boys and young men could be an indication that prevention efforts, including that of PrEP uptake by young men has been low. Additionally, emphasis is often placed on girls and young women and MSM [14, 15]. Adolescent boys also need customized HIV prevention [16] including understanding which HIV prevention methods they are comfortable with and want to use.

In this study, we used a mixed-methods approach to investigate PrEP preference and factors associated with on-demand PrEP preference among heterosexual adolescent boys and

young men in SSA (ABYM). Results from this analysis could help inform PrEP delivery policy strategies in this population.

## Materials and methods

### Study design

This study was part of tan HIV prevention programme called "The Combined HIV Adolescent Prevention Study (CHAPS), a joint African-European HIV prevention collaboration running in South Africa, Uganda, and Zimbabwe, focusing on young adults (13–24 years old) between April and November 2019. CHAPS comprised of two primary studies running sequentially: (i) Firstly, a social science mixed methods study investigating the acceptability and feasibility of taking daily and on-demand PrEP [17], and (ii) a Phase II randomized clinical trial investigating the optimum on-demand PrEP dosing schedule for insertive sex for both FTC-TDF and FTC-TAF (ClinicalTrials.gov NCT03986970, June 2019). Details of the CHAPS study have been previously described [18]. We used a cross-sectional sequential exploratory mixed-methods analysis of the data drawn from the CHAPS social science study.

### Setting, recruitment and eligibility

Purposive community-based sampling was used to recruit young people aged 13–24 years in four settings: Johannesburg and Cape Town in South Africa, Wakiso in Uganda, and Chitungwiza in Zimbabwe. In South Africa and Zimbabwe, participants were recruited from community groups, schools, churches whose HIV prevalence is 15% amongst 15–49 year olds, 24%, 20% amongst 15–24 year olds and 7.3% amongst 20–24 respectively [18], bars, taxi ranks, and other public meeting places, with CHAPS fieldworkers providing an overview of the study to establish initial interest. In Uganda, participants were recruited at fish landing sites, with information on the study being provided through local leaders, Village Health Teams, and project mobilisers.

Young people were eligible to participate in the survey if they were aged 13–24 years, sexually active (engaged in sexual intercourse in the past six months), willing to undergo HIV testing, and had no history of psychological/psychiatric disorder. Parental waiver of consent was approved by ethical boards for use in Cape Town, Entebbe, and Chitungwiza. Written study informed consent or assent was obtained from all participants.

### Data collection

The quantitative data were collected on tablets using Open Data Kit software, by trained research assistants. Data included PrEP preferences, reasons for preference, age, household details, frequency of sex in last month and hours of premeditation before sex, considering the most recent sexual encounter, a participant was asked to estimate how far ahead he was likely to have sex (premeditation about sex). This was collected as a categorical variable (<2 hrs, 2–12 hrs, 13-24hrs, >24hrs, prefer not to answer). More details on how this was phrased have been provided in the data analysis section of the manuscript. The timelines 2-24hours aligned with dosing recommendations for on demand PrEP [9].

Qualitative interviews were conducted by experienced social science researchers, who conducted group discussions (GDs) and In-Depth Interviews (IDIs), using a semi-structured topic guide, in a secure and private location that was comfortable for the interviewer and the interviewee. The place used was either suggested by the interviewee or pre-set by the interviewer at the local health clinic or community hall. The IDIs and GDs generally focused on motivations and hindrances for HIV testing, PrEP preference (daily vs on-demand), and

reasons for the PrEP preference. Data collection was conducted in a language comfortable to the participant, audio-recorded, transcribed verbatim, and later translated into English.

## Statistical analysis

We used STATA version 15 for cleaning and analysis of quantitative data. Data was collected on social demographic factors; age, household size, household headship, currently in school and sexual behaviour; frequency of having sex in the last one month, hours of premeditation about sex before the actual sexual encounter (<2 hours, 2–12, 13–24, >24 hours and prefer not to answer), condom use at the last sexual encounter (never, sometimes (<weekly), frequently (weekly/daily). Data on the main outcome were collected as a binary variable (participants chose between on-demand and daily oral PrEP).

Participant socio-demographic, sexual risk behaviour, and other variables were summarized using means with standard deviations or medians with interquartile range for continuous variables and frequencies with percentages for categorical variables, all stratified by country. Participant characteristics were compared between those who preferred on-demand and those that preferred daily oral PrEP using Chi-square tests. Logistic regression models were fitted to estimate factors associated with on-demand vs daily oral PrEP preference. Bivariable models were fitted first to determine unadjusted factors associated with on-demand PrEP preference. At bivariable analysis, variables that had a likelihood ratio test p-value of <0.150 (not Wald p-value for individual categories of the variable) were considered for multivariable analysis. The multivariable model was built basing on likelihood ratio test (LRT) p-value i.e. LRT p-value of addition into the model of a given factor. If a factor had a model p-value <0.05 at bivariable analysis but its LRT p-value for inclusion into the multivariable model was >0.05, that factor was excluded.

For highly correlated variables, we retained the one that was more statistically significantly associated with on-demand PrEP at unadjusted analysis basing on LRT p-value. Reasons for the choice of either mode of PrEP delivery were summarized using counts and percentages.

A thematic analysis approach was used to identify, analyse, and interpret patterns of meaning ('themes') within the qualitative data [19]. To attain this analysis, data familiarization was conducted, the researchers at each study site read the transcripts several times and made notes of key ideas and recurrent codes. Sampled transcripts were initially coded to identify emerging and recurrent themes which were compared across sites to ensure consistency and refine the coding framework and codebook. It is at this point that all the remaining transcripts were coded using the refined codebook. Data were then indexed by identifying segments of the data that corresponded to a particular code. The research teams discussed drawing and clarification of the data to develop broader categories allowing them to refine the meaning of the data in context according to each research question.

## Ethics approval

This study was conducted in line with the principles of the Declaration of Helsinki. Approval was granted by the Uganda Virus Research Institute Research and Ethics Committee (March 13, 2018; GC127/18/3/638); Uganda National Council for Science and Technology (June 18, 2018; SS 4579); the Joint Research Ethics Committee for the University of Zimbabwe, College of Health Sciences and the Parirenyatwa Group of Hospitals (JREC) (October 1, 2018; JREC/ 195/18); the Medical Research Council of Zimbabwe (MRCZ) and the Research Council of Zimbabwe (December 8, 2018; MRCZ/A/2356); the University of Cape Town Human Research Ethics Committee (August 8, 2018; 290/2018); and the London School of Hygiene and Tropical Medicine (October 26, 2018, 15629).

Prior to study participation, written informed consent was obtained from participants 18 years and older. Parental consent and participant assent were also required for participants 17 years and younger. Sites in Uganda, Zimbabwe, and Cape Town, South Africa had parental consent waivers in place according to their respective guidelines. Participants were reimbursed for their time and participation based on site national requirements and guidelines.

## Results

In total, 673 participants were enrolled. Of these, 26 (0.3%) participants were excluded from the analysis because they were either not sure or indifferent about their PrEP regimen preference. Of the remaining 647 participants, 422 (65.2%) preferred on-demand PrEP and 225 (34.8%) preferred daily PrEP.

### Participants' characteristics

In each country, the median age of the participants was 19 years, with about 71.7% of participants aged 18 to 24 years (Table 1). About half (52.6) were currently in school, a slightly higher proportion of 58.6% in South Africa than the other countries (Table 1). More participants reported heading their household in Uganda (38.0%) than in South Africa (4.5%) or Zimbabwe (10.1%). The proportion of participants reporting less than 2 hours premeditation about sex before the actual sexual encounter was higher in South Africa (36.9%) and Zimbabwe (28.6%) than in Uganda (18.8%). However, a higher proportion of participants in Uganda (30.4%) and Zimbabwe (25.6%) preferred not to reveal their time of premeditation about sex before actual encounter (Table 1).

### On-demand PrEP preference

The overall proportion of participants who stated a preference for on-demand PrEP was 422/647 (65.2%). There are differences between our survey participants from the different countries, Uganda (76.8%) and Zimbabwe (70.4%) than from South Africa (45.5%), p<0.001 (Table 2). The preference for on-demand PrEP increased with age; 13–15 years (47.9%), 16–17 years (60.7%) and 18–24 years (69.0%).

### Factors associated with on-demand PrEP (multivariable analysis)

Factors that were independently associated with preference for on-demand PrEP included country of residence [South Africa, adjusted odds ratio (aOR) = 0.2, 95% CI: 0.1–0.3, Zimbabwe aOR = 0.7, 9%CI 0.5–1.1) compared to Uganda]; hours of premeditation about sex which showed the odds of preference for on-demand PrEP was lower in participants that preferred not to answer this question aOR = 0.5, 95%CI: 0.3–0.9 compared to <2 hours (Table 2).

### Primary reasons for choosing on-demand PrEP

Among those who preferred on-demand PrEP, 38% cited not liking daily tablets, 17% thought they are not exposed to HIV frequently enough to warrant taking PrEP every day, 15% believed taking PrEP every day might make others think they have HIV and 11% thought fewer tablets meant lower pill burden or less pill fatigue (Table 3). Preference for daily PrEP was largely because of daily administration providing continuous protection against HIV (45%), more protection overall (19%), and a daily tablet providing more routine to administering PrEP rather than having to remember taking a tablet before and after) sex (15%).

**Table 1. Demographic and socio-behavioural characteristics stratified by country and PrEP preference for HIV at risk adolescent boys and young men enrolled in a multi-site study (Apr-Sep 2019).**

| | | Uganda, n = 250 | South Africa, n = 198 | Zimbabwe, n = 199 |
|---|---|---|---|---|
| Characteristics | | | | |
| Age (mean ±SD) | | 19.6±3.0 | 19.1±2.7 | 19.4±3.0 |
| Age, years | | | | |
| | 13–15 | 25(10.0%) | 26(13.1%) | 20(10.1%) |
| | 16–17 | 43(17.2%) | 29(14.7%) | 40(20.1%) |
| | 18–24 | 182(72.8%) | 143(72.2%) | 139(69.8%) |
| Currently Student | | | | |
| | Yes | 124(49.6%) | 116(58.6%) | 100(50.3%) |
| | No | 126(50.4%) | 82(41.4%) | 99(49.7%) |
| Household size (including respondent), categorized | | | | |
| | 1–5 | 240(96.0%) | 173(87.4%) | 189(94.9%) |
| | 6–20 | 10(4.0%) | 25(12.6%) | 10(5.1%) |
| Participant is household head | | | | |
| | Yes | 95(38.0%) | 9(4.5%) | 20(10.1%) |
| | No | 155(62.0%) | 189(95.5%) | 179(89.9%) |
| Frequency of sex in last month | | | | |
| | Every day | 2(0.8%) | 13(6.5%) | 3(1.5%) |
| | 2–3 times a week | 27(10.8%) | 56(28.4%) | 28(14.1%) |
| | Once a week | 20(8.0%) | 47(23.7%) | 30(15.1%) |
| | Once a month | 31(12.4%) | 35(17.7%) | 39(19.6%) |
| | Never | 87(34.8%) | 29(14.6%) | 48(24.1%) |
| | Prefer not to answer | 83(33.2%) | 18(9.1%) | 51(25.6%) |
| Hours of premeditation about sex (before actual sexual encounter) | | | | |
| | <2 hours | 47(18.8%) | 73(36.8%) | 57(28.7%) |
| | 2–12 hours | 26(10.4%) | 60(30.3%) | 32(16.1%) |
| | 13–24 hours | 26(10.4%) | 20(10.2%) | 8(4.0%) |
| | >24 hours | 75(30.0%) | 45(22.7%) | 51(25.6%) |
| | Prefer not to answer | 76(30.4%) | 0(0.0%) | 51(25.6%) |
| Condom use at the last sexual encounter | | | | |
| | Yes | 64(36.8%) | 122(61.6%) | 85(57.4%) |
| | No | 110(63.2%) | 76(38.4%) | 63(42.6%) |
| Alcohol use (6 or more drinks at a time) | | | | |
| | Never | 44(61.9%) | 34(19.6%) | 16(17.9%) |
| | Sometimes | 17(23.9%) | 103(59.6%) | 55(61.9%) |
| | Frequently | 10(14.2) | 36(20.8%) | 18(20.2%) |
| Number of sex partners in the past 6 months | | | | |
| | 0 | 2(1.0%) | 111(44.4%) | 68(34.2%) |
| | 1–3 | 146(73.7%) | 114(45.6%) | 107(53.8%) |
| | 4–9 | 36(18.2%) | 13(5.2%) | 20(10.0%) |
| | > = 10 | 5(2.5%) | 12(4.8%) | 4(2.0%) |
| | Don't remember | 9(4.6%) | 0(0.0%) | 0(0.0%) |

The above findings are supported by data from the qualitative component that showed that most of the young males preferred on-demand PrEP. From the qualitative data we identified these major themes.

**Table 2. Unadjusted and adjusted logistic regression models of the factors associated with choosing on-demand PrEP N = 647.**

| Variables | | n(%) | On demand | uOR (95% CI)* | Lrt (p-value) | aOR(95%)** | P-value |
|---|---|---|---|---|---|---|---|
| Country | | | | | <0.001 | | <0.001 |
| | Uganda | 250(38.6) | 192(76.8) | 1.0 | | 1.0 | |
| | South Africa | 198(30.1) | 90(45.5) | 0.3[0.2–0.4] | | 0.2[0.1–0.3] | |
| | Zimbabwe | 199(30.7) | 140(70.4) | 0.7[0.5–1.1] | | 0.7[0.5–1.1] | |
| Age, median (IQR) | | 19.4(17–22) | 18.7(16–21) | | | | |
| Age | | | | | 0.002 | | 0.002 |
| | 13–15 | 71(11.0) | 34(47.9) | 1.0 | | 1.0 | |
| | 16–17 | 112(17.3) | 68(60.7) | 1.7[0.9–3.1] | | 1.3[0.7–2.4] | |
| | 18–24 | 464(71.7) | 320(69.0) | 2.4[1.5–4.0] | | 1.6[0.8–2.8] | |
| Currently Student | | | | | 0.263 | | |
| | **Yes** | 340(52.6) | 215(63.2) | 1.0 | | | |
| | **No** | 307(47.5) | 207(67.4) | 1.2[0.9–1.7] | | | |
| Number of people in household | | | | | 0.003 | | |
| | 1–5 | 602(93.0) | 402(66.8) | 1.0 | | | |
| | 6–20 | 45(7.0) | 20(44.4) | 0.4[0.2–0.7] | | | |
| Heads household | | | | | <0.001 | | |
| | No | 523(80.8) | 322(61.6) | 1.0 | | | |
| | yes | 124(19.2) | 100(80.6) | 2.6[1.6–4.2] | | | |
| Frequency of sex in last month | | | | | <0.001 | | |
| | Never | 164(25.4) | 132(80.5) | 1.0 | | | |
| | Twice a day | 6(0.9) | 1(16.7) | 0.5[0.1–0.4] | | | |
| | Every day | 12(1.9) | 5(41.7) | 0.2[0.1–0.6] | | | |
| | 2–3 times a week | 111(17.2) | 66(59.5) | 0.4[0.2–0.6] | | | |
| | Once a week | 97(15.0) | 58(59.8) | 0.4[0.2–0.6] | | | |
| | Once a month | 105(16.2) | 72(68.6) | 0.5[0.3–0.9] | | | |
| | Prefer not to answer | 152(23.5) | 88(57.8) | 0.3[0.2–0.6] | | | |
| Hours of premeditation about sex (before actual sexual encounter) | | | | | 0.019 | | 0.013 |
| | <2 hours | 177(27.4) | 110(62.2) | 1.0 | | 1.0 | |
| | 2–12 hours | 118(18.2) | 71(60.2) | 0.9[0.6–1.5] | | 1.0[0.6–1.7] | |
| | 13–24 hours | 54(8.4) | 39(72.2) | 1.6[0.8–3.1] | | 1.5[0.7–3.1] | |
| | >24 hours | 171(26.4) | 127(74.3) | 1.8[1.1–2.8] | | 1.4[0.9–2.3] | |
| | Prefer not to answer | 127(19.6) | 75(59.1) | 0.9[0.6–1.4] | | 0.5[0.3–0.9] | |
| Condom use at the last sexual encounter | | | | | 0.002 | | |
| | No | 249(47.9) | 164(60.5) | 1 | | | |
| | Yes | 271(52.1) | 183(73.5) | 0.6[0.4–0.8] | | | |
| Alcohol use (6 or more drinks at a time) | | | | | 0.682 | | |
| | Never | 94(28.2) | 59(62.8) | 1 | | | |
| | sometimes | 175(52.6) | 112(64.0) | 1.1[0.6–1.8] | | | |
| | Frequently | 64(19.2) | 37(57.8) | 0.8[0.4–1.6] | | | |

OR = odds ratio; uOR = unadjusted odds ratio; aOR = adjusted odds-ratio; 95% CI = 95% confidence interval

## On-demand themes

**1) Concerns of taking daily pills.** The majority of male participants expressed the idea that taking daily pills was not convenient for them with potential challenges in adherence. They expressed concerns around failing to take the daily medication. This meant that they were not in support of taking pills if they had no planned sexual encounters. The dislike of

**Table 3. Main reasons associated with PREP preference among at-risk HIV adolescents.**

| Reasons for PrEP delivery mode preference | | | |
|---|---|---|---|
| **On-demand** | **n = 422 (65.2%)** | Daily | **n = 225 (34.8%)** |
| **I don't like taking tablets daily** | 158(37.5%) | All time protection, therefore, no need to have planned sex | 102(45.3%) |
| **Not at risk most of the time, therefore no need for PrEP everyday** | 71(16.8%) | Daily PrEP gives more protection than on-demand PrEP | 43(19.1%) |
| **Taking PrEP everyday may make people think that I have HIV** | 63(14.9%) | Because of unplanned sex, on-demand PrEP would be difficult | 26(11.5%) |
| **Fewer tablets means lower pill burden or less pill fatigue** | 45(10.7%) | Daily tablets easier to remember than PrEP around time of sex | 33(14.7%) |
| **Cheaper than taking everyday** | 25(5.9%) | At risk most of the time | 15(6.7%) |
| **Less tablets therefore easy to store** | 18(4.3%) | Reduce the chance of side effects | 6(2.7%) |
| **Other\*** | 41(9.7%) | Other | |
| **Not sure** | 1(0.2%) | Not sure | |

taking medication was associated with a preference for on-demand PrEP as expressed in the following quotes.

> *Because I don't want to take tablets all the time, yet you only take on demand when you are going to have sexual intercourse. The habit of taking drugs daily I don't like it and I even fear it. IDI Male 15 Years Uganda.*

> *So many young people do not want to take pills regularly. I see even at school many do not want to take pills even when they are ill. IDI Male 21 years Zimbabwe*

**2) I am not at risk most of the time so would not need PrEP every day.** Some participants reported that on-demand PrEP would be preferable as they were not at risk of contracting HIV most of the time and therefore PrEP would only be taken when they felt they are at increased risk as exemplified below:

> *The fact that am not someone who likes partying that I should expect three girls to sleep with per week, that's why I will only take on-demand well knowing that in two days am going to expect someone than taking it daily because then I will get tired. The on-demand PrEP is a better option. GD Males 13–17 years Uganda.*

> *"Looking at our community, everybody is always up and down, here and there, most of the people wouldn't like carrying their pills each day so I think taking PrEP every day when sex is not an everyday thing, I don't think that it's convenient, that is where on-demand becomes better." GD Males 18–24 years, South Africa*

Some participants were either not sexually active, not engaging in regular sexual activity or they were faithful to their partners. This made taking pills daily not applicable to them and therefore choosing to take on-demand PrEP when they felt they needed it. This was particularly the case when they thought they were going to be exposed to HIV.

> *it is not a must that you should be having sex with women all the time. I don't think whether it is possible because you might find that you may take it twice or thrice in a month. That day I take my pill, is the day I will have sex and it ends. That is better because when you get to*

*know that you have a match [sexual intercourse] before you call a girl to give her the programme you will first take your PrEP. IDI Male 22 years Uganda*

*Since you are not engaging sexually every day you may end up seeing daily PrEP as an unnecessary burden. . . A person may choose not to take daily PrEP because he/she can spend like three months without having sex. If a person reaches a time of having sex, he will remember to take it on demand. IDI Male 22 years Zimbabwe*

**3) Fewer tablets meant less pill burden or less pill fatigue.** Other participants reported other benefits of taking fewer tablets such as fewer pills meant fewer side effects as well as less tablet fatigue.

*Taking on-demand, will be easy for me because I will not be taking them every day it has to be on some occasions, I would prefer to take it on demand rather than daily. Pills do affect me whenever I take them. IDI Male 18 years, Uganda.*

*Some people have their illnesses, you may find one who takes drugs all the time because he could be having pressure or diabetes. So, you if add him PrEP, it may be a burden to him. One should take PrEP for just a year, and she stops. If a person takes PrEP yet takes other drugs say like, for Pressure and diabetes, he may get tired. IDI male 22 years Uganda*

*Taking pills every day, aaahh NO! I think we have the same mind-set because the ones I interact with them or when you have your friend, you probably think alike. So, I think this business of taking pills everyday uhmm. . .it's a NO NO! and for the case of on-demand they will be less pills. IDI Male 16 years Zimbabwe*

*If only there was a way of reducing the number to once after 2 months or once a month as Toni was saying, it would be better. That's what I think. Because a person of my age couldn't handle taking pills daily like an HIV person. GD male 15–17 years zimbabwe.*

**4) Taking PrEP every day may make people think that I have HIV.** Qualitative interviews showed that some adolescents refrained from taking daily PrEP because of its association with antiretroviral medications, and they did not want to be associated with people living with HIV, as expressed in the narrative below.

*I also see the same case because if I take it every day there will be no difference with an HIV positive person. People will start saying that you are HIV positive because you will be moving with your drugs at work, and they see you taking them. . .Male GD 19–24 Uganda.*

*Hmmm so that people may see you swallow prep daily. They start thinking about you differently, that you could be HIV positive" she is swallowing tablets but not getting cured, what is the matter." IDI Male 13 years Uganda.*

*There is a line for HIV services like to collect HIV pills, if ever they see you on that line, they would think that you are taking tablets. Sometimes this thing has to be private 'cause people do not feel free to get into the line, that is why some people do not take tablets and do not attend the clinic because everyone knows that this line is for HIV. They rather mix this line with the people who are suffering from headaches, it should be a mix. IDI Male 17 years South Africa"*

Some participants also feared taking daily tablets would make others perceived them as being unwell, as narrated in the excerpts below.

*"Because I think they don't wanna get judged by people that they are taking PrEP." I think others will think that if they take PrEP, they will think that they are sick or something else."*
*GD male 13–17 South Africa*

*I would not be comfortable because people will think I am sick. . . IDI Male 13 years Zimbabwe.*

### Daily PrEP themes

The factors that determined the need/preference for daily PrEP were also explored among young male participants. Their narratives revealed several reasons why they thought daily PrEP was the best option as opposed to on-demand.

Reasons for participants preferring daily PrEP included being perceived as more effective than on-demand PrEP, and frequency of sexual intercourse necessitating daily use. Participants' responses were categorized to the following themes:

**1) It provides protection all the time.**   Full-time protection was mostly noted by the young male participants. This meant that they would not need to waste time if an opportunity arose to engage in sexual activity as they would always be prepared, and also enable them to enjoy sex without a condom. Therefore, the participants who preferred daily PrEP believed that they would be more and better protected 24/7, hence the preference for daily PrEP linked to feeling safer and protected everyday against acquiring HIV ignoring other concerns such as STIs and unintended pregnancy.

*I think it will be easy to take PrEP every day because you will know that you don't want to be infected by HIV. So, the moment that you decide that you don't want HIV it will simply become easy for you. GD Male 13–15 years Zimbabwe*

*Taking PrEP when required is not ideal because you may be involved in an emergency, and you end up having sex without taking your medication so taking PrEP every day you are always protected which is right. GD male 18–24 years South Africa.*

*On the other side it's ok depending on your situation and how you are living your life, are you careless and you are at risk of getting infected, then you need to take the pill daily. IDI Male 18 years, Zimbabwe.*

*Every day I am meeting new people and as I said before, there are lustful feelings so it is going to be wise and very helpful for me to take it [PrEP] every day actually so that whoever I meet along the way I would know I am safe. IDI Male 22 years, South Africa*

**2) I do not plan sex therefore on-demand PrEP would be difficult to take.**   Some participants mentioned not being able to predict sexual activity as another reason for daily PrEP preference. The participants revealed that they never plan when to have sex, therefore it would be better to be protected all the time through daily PrEP.

*I don't support on demand PrEP because you might reach a time when you are at risk yet you did not move with the medicine or when you are not ready. If you have been taking daily PrEP, you can be sure of the protection even when you are going to have sex. GD Male 13–17 years Uganda*

*I think taking PrEP daily is better than on demand because there are sometimes you end up having sex without prior planning to have it and you wouldn't have taken your medication.*

*As a young person, I may have feelings and have unprotected sex when we have not planned it but because I am taking daily PrEP that way I will be safe from contracting the virus. IDI 20 years Zimbabwe.*

*Yah I would prefer taking PrEP daily. You never know coz sex is not something that you plan, it just happens so if it is something that we don't plan then I will take PrEP daily just to be safe. You just have to be prepared for anything always instead of being caught off guard IDI 21 Male South Africa*

**3) I am at risk most of the time so I would need PrEP every day.** Besides the stigma related to daily drugs, participants also perceived daily PrEP as a more effective way to continue high risk sex. They confirmed the use of daily PrEP associated with self-perceived HIV risk and continuous engagement in and in high-risk behaviour. Therefore, PrEP would help to give them daily protection from HIV as narrated in the excerpts below.

*I would feel good and comfortable because I like enjoying my life and having sex without a condom with different girls, I am okay with that. That is why I support the daily PrEP because it is better to take daily, and you are prepared when you are going to have sex. The one that is taken on-demand when you are going to have sex, you may not have it or easily get it when you are ready to have sex and it is too late. IDI Male 18 yeas Uganda*

*I want to take it every day because my friends say we are at risk anytime. They can give you a drink without yourself knowing it has been laced with drugs and you end up having sex so it's wise to be always protected GD male 13–15 years Zimbabwe.*

**4) I like the routine of daily tablets rather than remembering PrEP just at sex time.** From the interviews, some participants acknowledged the possibility of only remembering to take on-demand PrEP at the start of sex, given the spontaneity of sex in the heat of the moment making it too late to take PrEP and therefore affecting sexual intimacy. Additionally, some participants with the habit of taking drugs regularly mentioned this would favor them to take daily PrEP.

*When you want to have sex, you want it then so there is no time to start thinking of taking a pill first. You cannot tell you partner, hey, stop just wait a minute and I take my PrEP. She may just lose it and you miss out. IDI Male 21 years Uganda.*

*Well it won't be difficult to take daily PrEP because I know I have to have breakfast every morning and my other medicines then take one PrEP pill every day it is not that hard plus it no of waste time it is less than a minute. IDI Male 20 years South Africa*

## Discussion

This paper presents analyses of PrEP preference among young heterosexual men from SSA and shows that PrEP was highly accepted by this population We found that nearly two thirds of young men preferred on-demand PrEP with considerable inter-country variability which is in line with previous study findings in the same age group [20, 21].

The three top-rated reasons for which adolescents preferred on-demand PrEP were disliking taking daily tablets (37.5%), thinking they are not at frequent risk of HIV (16.8%) and thinking that taking PrEP every day would make people think that they have HIV (14.9%). Given the dislike for taking daily tablets was more than double any other reason for choosing

on-demand PrEP, highlights pill burden as a barrier for adolescents and young men, and the need to have on-demand options. A significant number of young men planned their sexual activity thus allowing them to plan their on-demand PrEP uptake would be appropriate. This contrasts with young females/adolescents who may be less empowered to plan their sexual activity.

The top-rated reasons for preferring daily PrEP were participants wanting protection all the time so that they do not need to plan when to have sex (45.3%), believing daily PrEP is more effective than on-demand PrEP (19.1%) and for those who did not plan sex concern that they would not be able to take on-demand PrEP accurately (11.5%). These findings underpin the need for educating young men on the effectiveness of both PrEP regimens, the pros and cons of each method according to sexual behavioural practices, as well as ensuring they know PrEP only provides protection against HIV, and that risks of unprotected sexual intercourse other than HIV acquisition are of other sexually transmitted infections and pregnancy.

The main strength of this study is that it is the first multi-country study in SSA focussing on adolescent boys PrEP preference and reasons for preference, a neglected group unlike their counterparts in the field of HIV prevention

Even though the WHO includes adolescents as a priority group for PrEP [4], our limitation is that we were asking about hypothetical use of PrEP rather than assessing actual use and reasons for use. Another limitation is that even though long-acting injectable Cabotegravir may be offered as an additional prevention choice for people at substantial risk of HIV infection, as part of combination prevention approaches [22] in this study we did not provide it as a choice for the adolescents because it was only approved in late 2022, and was yet to be rolled out in healthcare facilities. Participants were sampled purposively for inclusion in the study, therefore our findings may not be generalisable to all young men in these settings.

Future research is required among adolescent boys in SSA who have real-life experience with both on-demand and daily PrEP to understand their choices and experiences. Longitudinal studies will facilitate understanding of how adolescent boys' PrEP preferences change over time and how willing they are to switch between regimens. We hope the study will encourage healthcare providers to support PrEP uptake in young men and offer them both daily and on-demand PrEP options.

## Conclusion

In the study, we found high acceptance of on demand PrEP among adolescent and young heterosexual men, particularly in Uganda and Zimbabwe. Following WHO recommendation for on-demand (event-driven) oral PrEP for MSM [12],Our findings present an opportunity to engage regional policy makers and WHO about including on-demand strategies for young heterosexual men in PrEP programming similar to MSMs [12] and provide some insights into their HIV prevention decision making.

We hope that given our results, stakeholders can take into account that there is high user acceptability of this form of HIV prevention.

## Acknowledgments

We acknowledge the contributions of the ITAPS team, CHAPS study team and all the participants.

## Author Contributions

**Conceptualization:** Ayoub Kakande, Emily Webb.

**Data curation:** Andrew Abaasa.

**Formal analysis:** Ayoub Kakande, Andrew Abaasa, Emily Webb.

**Investigation:** Helen Anne Weiss, Emily Webb, Janet Seeley, Julie Fox.

**Software:** Ayoub Kakande.

**Supervision:** Ayoub Kakande.

**Validation:** Ayoub Kakande.

**Visualization:** Ayoub Kakande.

**Writing – original draft:** Ayoub Kakande, Lynda Stranix-Chibanda.

**Writing – review & editing:** Ayoub Kakande, Andrew Sentoogo Ssemata, Richard Muhumuza, Millicent Atujuna, Andrew Abaasa, Denis Ndekezi, Gugulethu Tshabalala, Teacler Nematadzira, Stefanie Hornschuh, Mangxilana Nomvuyo, Nadia Ahmed, Mamakiri Maluadzi, Helen Anne Weiss, Emily Webb, Janan Janine Dietrich, Janet Seeley, Julie Fox.

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
