## [Decision Letter · Decision Letter 0]

6 Dec 2022

PONE-D-22-28042PrEP preference among adolescent boys and young men in three sub-Saharan African countriesPLOS ONE

Dear Dr. Kakande,

Thank you for submitting your manuscript to PLOS ONE. After careful consideration, we feel that it has merit but does not fully meet PLOS ONE’s publication criteria as it currently stands. Therefore, we invite you to submit a revised version of the manuscript that addresses the points raised during the review process.

We look forward to receiving your revised manuscript.

Kind regards,

Joseph KB Matovu, Ph.D.

Academic Editor

PLOS ONE

Journal Requirements:

Reviewers' comments:

Reviewer's Responses to Questions

**Comments to the Author**

1. Is the manuscript technically sound, and do the data support the conclusions?

Reviewer #1: Yes

Reviewer #2: Partly

2. Has the statistical analysis been performed appropriately and rigorously? 

Reviewer #1: Yes

Reviewer #2: No

3. Have the authors made all data underlying the findings in their manuscript fully available?

Reviewer #1: Yes

Reviewer #2: Yes

4. Is the manuscript presented in an intelligible fashion and written in standard English?

Reviewer #1: Yes

Reviewer #2: Yes

5. Review Comments to the Author

Reviewer #1: 1. ETHICAL CONSIDERATIONS: Why was the consent not sought for the younger boys? Was assent sought?

2. The authors can give slightly more contextual background information about the adolescents from the 3 different countries...to give more insight in the study findings

3. Purposive community-based sampling data collection methodology worked well for the study but replication of the study design is limited.

Reviewer #2: This manuscript explores preferences for on-demand vs. daily PrEP among adolescent boys and young men in SSA, a group at high risk of HIV infection. Overall, I found the qualitative data compelling and interesting. I think there are several issues regarding framing of on-demand PrEP and the quantitative analysis that need to be addressed. First, to the best of my knowledge, on-demand PrEP has been formally recommended by public health organizations for heterosexual males. This may not be true, but the authors do not provide citations supporting on-demand PrEP for heterosexual males. This is not a major limitation, but I think their study needs to be better contextualized in the Intro and Discussion as on-demand PrEP would be “off-label” for most participants, and has only been studied with TDF-FTC, not TAF-FTC. Unless on-demand PrEP has been formally recommended for heterosexual men, I think conclusions from this study need to be better contextualized. Second, I think the multivariable model needs to be clarified, which variables were included in the final adjusted model? The quantitative results are confusing as well, when the one group found to be significantly associated with preferring on-demand PrEP is “prefer not to answer,” I don’t know what to make of this finding. Third, I think greater detail to the mixed methods strategy is needed: was this a primarily explanatory model vs exploratory? It seems the data were obtained sequentially but analyzed simultaneously, but it is not clear.

1.Title: After reading the abstract, it becomes clear that the focus of the research is on-demand vs. daily PrEP. I would suggest including this focus in the title, since “PrEP preference” is a bit vague.

2.Abstract: Authors should specify mixed methods methodology briefly in abstract: was the process sequential (first quantitative, then qualitative?) or simultaneous? Exploratory vs. explanatory? I would also mention that these data came from a larger study… did the social science study precede the RCT?

3.Abstract: The factor “advanced planning of sex” needs to be clarified in the Results. This is confusing since the group is “prefer not to answer.”

4.Introduction, third paragraph: Since long acting injectable PrEP is now available in many regions, the authors should clarify PrEP as oral PrEP. The IPERGAY study also only considered TDF/FTC.

5.Introduction, third paragraph, last sentence: this sentence and the citations are misleading. On-demand PrEP is not recommended in that citation provided (from 2015); I believe it was not recommended by the WHO formally until 2019, and even so, was done with caveats: only for men who have sex with men. While I don’t think this is a major weakness of the paper, this caveat needs to be recognized earlier (the authors do mention in the third paragraph, but I think it needs to be earlier), particularly since the focus of this paper is on heterosexual adolescent boys and young men.

6.Methods, Data collection: Would re phrase “reasons for the uptake of PrEP uptake.”

7.Methods: Did the authors collect data on number of partners? Or if they had a regular partner?

8.Methods, Data Analysis: The authors state that the main outcome was analyzed as a binary variable. Was it also collected as binary, or were participants given a Likert scale of preference for on-demand vs. daily PrEP?

9.Methods: Additional detail on study measures would be helpful. For instance, how was the measure of “hours of premeditation before sex” phrased?

10.Methods: The authors should include more details on the type of mixed methodology used, e.g., sequential vs. simultaneous? Exploratory?

11.Methods/Results: The authors chose a backward elimination model, yet the results in Table 2 are a little confusing. Some factors in the bivariate model (frequency of sex, heads household, number of people in the household) appear significant at the bivariate model, but are not included in the final model. It is also unclear whether age was significant in the adjusted model, since the 95% Cis cross 1, but the p-value is 0.002.

12.Results: The authors state that “odds of preference for on-demand PrEP increased with increased hours of premeditation about sex” … this is only true in the unadjusted model, correct? In addition, the relationship does not appear linear. I also do not know what to make of the “prefer not to answer” being significant… since this could be very little time, or a lot of time.

13.Results, Qualitative (page 15-16) I would suggest retitling point 3) since many of the quotes deal with tablet fatigue, rather than side effects. These are related but not quite the same.

14.Results: Would suggest moving Table 3 higher up in the Results, in the beginning of the qualitative themes.

15.Discussion: Some mention of long acting injectables deserves mention, perhaps as a limitation of the study if not discussed.

16.Conclusions: the authors state “following WHO support for on-demand PrEP for all men, not just MSM…” I am not sure this is correct, and the authors have not cited WHO literature supporting on-demand PrEP for heterosexual men.

6. PLOS authors have the option to publish the peer review history of their article (what does this mean?). If published, this will include your full peer review and any attached files.

Reviewer #1: **Yes: **dr lydia atambo

Reviewer #2: No

---

## [Author Response · Author response to Decision Letter 0]

6 Feb 2023

Reviewer #1: 1. ETHICAL CONSIDERATIONS: Why was the consent not sought for the younger boys? Was assent sought?

Response: “Parental waiver of consent was approved by ethical boards for use in Cape Town, Entebbe, and Chitungwiza. Written study informed consent or assent was obtained from all participants”. 

Prior to study participation, written informed consent was obtained from participants 18 years and older. Parental consent and participant assent were also required for participants 17 years and younger. Sites in Uganda, Zimbabwe, and Cape Town, South Africa had parental consent waivers in place according to their respective guidelines. Participants were reimbursed for their time and participation based on site national requirements and guidelines

. 

2. The authors can give slightly more contextual background information about the adolescents from the 3 different countries...to give more insight in the study findings

Response: In sub-Saharan Africa, the overall prevalence of ‘any substance use’ is estimated at 41.6% with caffeine containing products at 41.2% and alcohol at 32.8% calling for interventions that are comprehensive and targeted at adolescents these settings(6). Many adolescents are at risk of HIV because 

Purposive community-based sampling was used to recruit young people aged 13-24 years in four settings: Johannesburg and Cape Town in South Africa, Wakiso in Uganda, and Chitungwiza in Zimbabwe. In South Africa and Zimbabwe, participants were recruited from community groups, schools, churches whose HIV prevalence is 15% amongst 15-49 year olds, 24% , 20% amongst 15-24 year olds and 7.3% amongst 20-24 respectively(18), bars, taxi ranks, and other public meeting places, of multiple partnerships and insufficient condom(7).

3. Purposive community-based sampling data collection methodology worked well for the study but replication of the study design is limited.

Response: Thanks for the comments. We acknowledge the difficulty with replication of a study that used a non-probability sampling method such as purposive sampling. We have now included this as a limitation in the discussion section.

---

## [Decision Letter · Decision Letter 1]

27 Feb 2023

PONE-D-22-28042R1PrEP preference among adolescent boys and young men in three sub-Saharan African countriesPLOS ONE

Dear Dr. Kakande,

Thank you for submitting your manuscript to PLOS ONE. After careful consideration, we feel that it has merit but does not fully meet PLOS ONE’s publication criteria as it currently stands. Therefore, we invite you to submit a revised version of the manuscript that addresses the points raised during the review process.

We look forward to receiving your revised manuscript.

Kind regards,

Joseph KB Matovu, Ph.D.

Academic Editor

PLOS ONE

Journal Requirements:

Additional Editor Comments:

The authors should attend to the minor comments raised by one of the reviewers before a final decision is made on this manuscript.

Reviewers' comments:

Reviewer's Responses to Questions

**Comments to the Author**

1. If the authors have adequately addressed your comments raised in a previous round of review and you feel that this manuscript is now acceptable for publication, you may indicate that here to bypass the “Comments to the Author” section, enter your conflict of interest statement in the “Confidential to Editor” section, and submit your "Accept" recommendation.

Reviewer #1: All comments have been addressed

Reviewer #2: (No Response)

2. Is the manuscript technically sound, and do the data support the conclusions?

Reviewer #1: Yes

Reviewer #2: Partly

3. Has the statistical analysis been performed appropriately and rigorously? 

Reviewer #1: Yes

Reviewer #2: I Don't Know

4. Have the authors made all data underlying the findings in their manuscript fully available?

Reviewer #1: Yes

Reviewer #2: Yes

5. Is the manuscript presented in an intelligible fashion and written in standard English?

Reviewer #1: Yes

Reviewer #2: Yes

6. Review Comments to the Author

Reviewer #1: The feedback has been received and reviewed. The feedback is satisfactory.

I acknowledge and appreciate the other comments that were shared by the other reviewer. Once they have been addressed, the publication can be published.

Reviewer #2: I appreciate the author’s response and believe the manuscript has been strengthened. However, there are still issues that remain unaddressed in the revision, including, I believe significantly, a clarification in the title of the manuscript. Given the large literature on different PrEP preferences, further specificity would serve the authors and manuscript well that would increase the impact of this manuscript. All of these comments should be easily addressable.

Title: While I appreciate the author’s comment on why they retained the title, perhaps I need to clarify why I think “PrEP preference” is too vague. This reviewer believes it is imperative to clarify in the title what the authors are investigating, given the large literature on PrEP preferences. PrEP preference could mean mode of delivery (e.g., oral vs. injectable vs. implantable), rather than oral dosing schedule. Even if injectable is not approved yet in this region, since this manuscript only assesses “preferences” (and not use), it is reasonable to assume that the authors intend to study injectable PrEP from the title alone. Another alternative “preference” would be site of delivery, e.g., clinic based, home delivery, pharmacy delivery, however, the authors do not investigate this. I believe further clarity in the title is needed, as both a reviewer and a potential reader of the manuscript would desire further clarification of WHAT is exactly being studied. A suggested title revision that would address my concern would be “Preferences for on-demand oral PrEP among…” or “Preferences for oral PrEP dosing”

Introduction, line 83: suggest clarifying in this sentence “Oral pre-exposure prophylaxis”

Introduction, line 93 (and elsewhere): suggest changing “regime” to “regimen”

Results: If the number of partners was measured, I would include it in Table 1 as a descriptive characteristic. Reasonable to leave out of analysis.

Data Analysis: Include the primary outcome in the data analysis section. I’m not sure why this was not included in the revision.

Prior point about multivariable model building: The reviewer appreciates the authors explanation of how they chose to build the multivariable model, but is again confused why this explanation is not included in the revised manuscript.

Table 3: Does the column reading “Less tables mean fewer side effects” need to also be revised to “fewer tables mean lower pill burden or less pill fatigue?” as revised elsewhere in the manuscript?

7. PLOS authors have the option to publish the peer review history of their article (what does this mean?). If published, this will include your full peer review and any attached files.

Reviewer #1: No

Reviewer #2: No

---

## [Author Response · Author response to Decision Letter 1]

21 Mar 2023

20th Mar 2023

Dear Editor, 

Re: Response to the reviewers’ comments: Ref. No: [PONE-D-22-28042R1] - [EMID:f381a81dd41eee53]

Title: 

PrEP preference among adolescent boys and young men in three sub-Saharan African countries

Thank you very much for reviewing our manuscript and providing us the opportunity to respond to the reviewers’ comments. Please see below our responses to the issues raised by the reviewers. 

 Yours sincerely,

Ayoub Kakande

Submitting Author 

On behalf of the CHAPS consortium

 

6. Review Comments to the Author

Title: While I appreciate the author’s comment on why they retained the title, perhaps I need to clarify why I think “PrEP preference” is too vague. This reviewer believes it is imperative to clarify in the title what the authors are investigating, given the large literature on PrEP preferences. PrEP preference could mean mode of delivery (e.g., oral vs. injectable vs. implantable), rather than oral dosing schedule. Even if injectable is not approved yet in this region, since this manuscript only assesses “preferences” (and not use), it is reasonable to assume that the authors intend to study injectable PrEP from the title alone. Another alternative “preference” would be site of delivery, e.g., clinic based, home delivery, pharmacy delivery, however, the authors do not investigate this. I believe further clarity in the title is needed, as both a reviewer and a potential reader of the manuscript would desire further clarification of WHAT is exactly being studied. A suggested title revision that would address my concern would be “Preferences for on-demand oral PrEP among…” or “Preferences for oral PrEP dosing”

Response: We thank the reviewer for this comment, we have edited the title to “Preferences for oral PrEP dosing among adolescent boys and young men in three sub-Saharan African countries”

Introduction, line 83: suggest clarifying in this sentence “Oral pre-exposure prophylaxis”

Introduction, line 93 (and elsewhere): suggest changing “regime” to “regimen”

Response: We thank the reviewer for this comment, we have edited pre-exposure prophylaxis to read Oral pre-exposure prophylaxis, and we have also edited all regimes to regimens

Results: If the number of partners was measured, I would include it in Table 1 as a descriptive characteristic. Reasonable to leave out of analysis.

Response: We thank the reviewer for this comment, we have added “number of sex partners in the last 6 months in table 1”

Data Analysis: Include the primary outcome in the data analysis section. I’m not sure why this was not included in the revision.

Response: We thank the reviewer for this comment, in the data analysis section we have included “Data on the main outcome were collected as a binary variable (participants chose between on-demand and daily oral PrEP). 

Prior point about multivariable model building: The reviewer appreciates the authors explanation of how they chose to build the multivariable model, but is again confused why this explanation is not included in the revised manuscript.

Response: we thank the reviewer for this comment and acknoldge the omission, we have now included it as “At bivariable analysis, variables that had a likelihood ratio test p-value of <0.150 (not Wald p-value for individual categories of the variable) were considered for multivariable analysis. The multivariable model was built basing on likelihood ratio test (LRT) p-value i.e. LRT p-value of addition into the model of a given factor. If a factor had a model p-value <0.05 at bivariable analysis but its LRT p-value for inclusion into the multivariable model was >0.05, that factor was excluded.” 

Table 3: Does the column reading “Less tables mean fewer side effects” need to also be revised to “fewer tables mean lower pill burden or less pill fatigue?” as revised elsewhere in the manuscript? 

Response: We thank the reviewer for this comment, we have replaced “Less tablets mean fewer side effects” with “fewer tablets mean lower pill burden or less pill fatigue?

---

## [Decision Letter · Decision Letter 2]

16 Apr 2023

Preferences for oral PrEP dosing among adolescent boys and young men in three sub-Saharan African countries

PONE-D-22-28042R2

Dear Dr. Kakande,

We’re pleased to inform you that your manuscript has been judged scientifically suitable for publication and will be formally accepted for publication once it meets all outstanding technical requirements.

Kind regards,

Joseph KB Matovu, Ph.D.

Academic Editor

PLOS ONE

Additional Editor Comments (optional):

Reviewers' comments:

Reviewer's Responses to Questions

**Comments to the Author**

1. If the authors have adequately addressed your comments raised in a previous round of review and you feel that this manuscript is now acceptable for publication, you may indicate that here to bypass the “Comments to the Author” section, enter your conflict of interest statement in the “Confidential to Editor” section, and submit your "Accept" recommendation.

Reviewer #2: All comments have been addressed

2. Is the manuscript technically sound, and do the data support the conclusions?

Reviewer #2: Yes

3. Has the statistical analysis been performed appropriately and rigorously? 

Reviewer #2: Yes

4. Have the authors made all data underlying the findings in their manuscript fully available?

Reviewer #2: Yes

5. Is the manuscript presented in an intelligible fashion and written in standard English?

Reviewer #2: Yes

6. Review Comments to the Author

Reviewer #2: This reviewer thanks the authors for addressing their concerns and comments in this current revision. I have no further comments.

7. PLOS authors have the option to publish the peer review history of their article (what does this mean?). If published, this will include your full peer review and any attached files.

Reviewer #2: No

---

## [Editor Report · Acceptance letter]

24 May 2023

PONE-D-22-28042R2 

Preferences for oral PrEP dosing among adolescent boys and young men in three sub-Saharan African countries 

Dear Dr. Kakande:

I'm pleased to inform you that your manuscript has been deemed suitable for publication in PLOS ONE. Congratulations! Your manuscript is now with our production department. 

Kind regards, 

on behalf of

Dr. Joseph KB Matovu 

Academic Editor

PLOS ONE